# Multi-Object Tracking Algorithm of Fusing Trajectory Compensation

**Jianhai Jin** [1,2]**, Liming Wang** [3]**, Qi You** [3] **and Jun Sun** [3,]*

1 China Ship Scientific Research Center, No. 265 Shanshui East Road, Wuxi 214082, China; jjh@cssrc.com.cn
2 Taihu Laboratory of Deepsea Technology Science, Shanshui East Road, Wuxi 214082, China
3 School of Artificial Intelligence and Computer Science, Jiangnan University, Lihu Avenue,
  Wuxi 214122, China; 6191910033@stu.jiangnan.edu.cn (L.W.); 7171905005@stu.jiangnan.edu.cn (Q.Y.)
* Correspondence: junsun@jiangnan.edu.cn

**Abstract:** Multi-object tracking (MOT) is an important research topic in the field of computer vision, including object detection and data association. However, problems such as missed detection and trajectory mismatch often lead to missing target information, thus resulting in missed target tracking and trajectory fragmentation. Uniform tracking confidence is also not conducive to the full utilization of detection results. Considering these problems, we first propose a threshold separation strategy, which sets different tracking thresholds for similarity matching and intersection over union (IoU) matching during association to make the distribution of detection information more reasonable. Then, the missing trajectories are screened and compensated with the predicted trajectories to improve the long-term tracking ability of the algorithm. When applied to different association algorithms or tracking algorithms, a better improvement effect can be obtained. It can achieve high tracking speed while achieving high tracking accuracy on the MOT Challenge dataset.

**Keywords:** multi-object tracking; data association; trajectory compensation

**MSC:** 68T45

## 1. Introduction

The multi-object tracking algorithm is inseparable from the accurate positioning information of the target, so that accurate detection results can bring better tracking results. At present, the target detection algorithm shows great progress [1–5], but the detection effect is not always ideal due to the complex tracking environment. Therefore, the rational use of detection information is also the focus of this task [6]. Although current association algorithms can combine different types of information and use multiple methods for tracking [7], it is still difficult to make full use of detection information. These algorithms can only track the detected targets and cannot supplement the undetected targets. Moreover, offline tracking can supplement the missing information in the current trajectory based on subsequent trajectory information or overall trajectory information, but online tracking cannot predict the image content of the next frame. In order to enhance the robustness of tracking and prevent the detection performance from determining the upper limit of the tracking performance, some researchers have begun to study how to use the existing trajectory to compensate for the lack of detection.

One approach is to model historical trajectories to directly predict the trajectory of the current image in an end-to-end fashion [8,9]. Another approach is to use historical trajectory information to aid detection. For example, MOTDT [10] supplements the detection candidate box by predicting the position of the trajectory in the current image as a generation candidate box, and then completes the supplementation of detection information through a series of screening steps. TraDes [11] fuses the tracked information into the current frame to address detection occlusion issues. However, the end-to-end trajectory

modeling method lacks scalability, and supplementing the detection information alone cannot solve the information loss caused by trajectory mismatch.

In this paper, we analyze the reasons for the abnormal termination of the trajectory from the perspective of the trajectory itself. Since the movement of the target is an existing process, the normal trajectory termination occurs mainly because the target gradually disappears from the field of view, such as being obscured or moving out of the field of view until it disappears completely. In this process, the detection confidence should have a gradually decaying trend. When the trajectory with high confidence of the target matched last time is suddenly lost, it is most likely due to missing target information. Therefore, we propose a multi-object tracking algorithm of fusing trajectory compensation, named FTC.

FTC uses different thresholds to filter candidate targets for two matches during data association. Using different confidence thresholds in different matching processes can not only provide high-quality appearance features for similarity matching, but also can provide more candidate bounding boxes for IoU matching, making detection information allocation more reasonable. After completing two matches, FTC starts from the trajectory of the missing target, analyzes the trajectory of the unmatched target, and selects the missing trajectory from the active trajectory. Then, the missing trajectory is extended for several frames according to the predefined extension threshold and confidence decay coefficient to realize the compensation for the missing information.

Furthermore, we use YOLOX-s [5] to build a one-shot tracking model for experimental validation. The experimental results show that our method can effectively solve the problem of missing information and bring about a significant improvement in tracking performance. The effect of applying FTC on different association and tracking algorithms also demonstrates the scalability of our method.

The article proceeds as follows. Section 2 introduces the FTC algorithm. Section 3 describes the process of building an efficient one-shot multi-object tracking model for experimental validation. The experimental results and analysis are given in Section 4, including ablation experiments and comparisons with other algorithms. Section 5 concludes this paper.

## 2. FTC

As shown in Figure 1, FTC contains two improvements (marked by dotted lines). First, FTC sets different thresholds to filter the targets to be matched so as to fully exploit the detection information. Then, FTC uses the predicted trajectory position to expand the trajectory that is judged to be missing, so as to compensate for the missing information.

Considering the excellent prediction effect and wide application of the Kalman filter (KF) algorithm [12], we employ a Kalman filter to predict trajectory positions, and assume that the target is in gentle motion (i.e., no sudden movements and stops). The pseudocode of FTC is shown in Algorithm 1.

We take a video sequence $V$ as input, along with a detector with embeddings $Det$ and $KF$. There are five thresholds, including $T_{emb}$, $T_{iou}$, $T_{init}$, $T_{extd}$ and $\alpha$. $T_{emb}$, $T_{iou}$ and $T_{init}$ are tracking thresholds. $T_{extd}$ is a trajectory expansion threshold and $\alpha$ is a confidence decay factor. The output of FTC is the track $T$ of the video. Each track contains the bounding box and the identity. The confidence of the latest matching target is recorded as the current score of the trajectory, and the trajectory that does not lose the target in the last match is recorded as the active state.

For each frame of the input video, the predicted detection boxes, detection confidence and corresponding embedding features are simultaneously obtained through $Det$. Then, the target is divided into $D_{emb}$ and $D_{iou}$ according to different thresholds, and the trajectory state is predicted by using the Kalman filter, as shown in lines 4 to 8 of Algorithm 1. When associating, we first perform feature similarity matching based on the embeddings of $D_{emb}$ and the current trajectory $T$. Then, we update the matched target to $T$, and record the unmatched target to $D_{remain}$ and trajectory to $T_{remain}$, as shown in lines 9 to 11 of Algorithm 1. Unmatched targets $D_{remain}$ need to be merged into $D_{iou}$ before the second

match. The second match uses the IoU distance to match the unmatched trajectory $T_{remain}$ with the target $D_{iou}$, and updates the matched target to $T$. The unmatched target records to $D_{unuse}$ and the unmatched trajectory records to $T_{lost}$. $D_{iou}$ contains targets with low confidence. In order to prevent low-quality embeddings affecting subsequent trajectory matching, for matches with confidence less than $T_{emb}$, their embedding features are not updated to the trajectory. Considering the length of the algorithm, we do not describe it in detail in Algorithm 1.

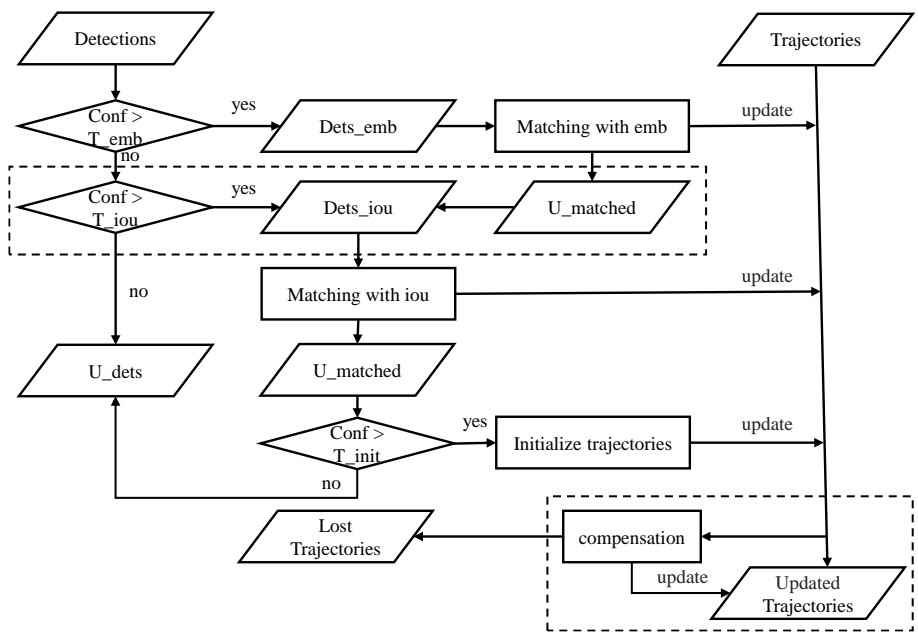

**Figure 1.** FTC.

Afterwards, the unmatched trajectories and targets are processed separately. First, we determine whether each unmatched trajectory is a missing trajectory. The judgment condition is in the 17th line of Algorithm 1. Three conditions must be satisfied at the same time—that is, the trajectory score is greater than the trajectory expansion threshold, the trajectory length is greater than 2, and the trajectory is active.

These three conditions are described in detail as follows. (1) Only if the trajectory score, i.e., the confidence of the latest matching target, is greater than a certain threshold will the possibility that the trajectory belongs to a sudden interruption be higher. (2) The trajectory with a length equal to 1 is the initial trajectory, and many initial trajectories are initialized from single-frame high-confidence targets generated by detection or association problems, and usually will not match new targets to become long trajectories. At this time, it is impossible to determine whether the trajectory is a real trajectory, so the initial trajectory is also called an undetermined trajectory. To prevent the expansion of undetermined trajectories, trajectories with a trajectory length less than 2 are excluded. (3) The activation state of the trajectory means that the trajectory has recently completed the matching of the new target in the previous frame. Trajectory expansion is not performed on trajectories that have not been updated twice or more.

When the track is judged to be a missing track, we update the predicted position of the track to the current track. Note that the update here is different from the previous two matching process updates. In the first two matches, the trajectory position predicted by the *KF* is the predicted value, and the matched target position is the detection result. When the trajectory is updated, the Kalman filter will use the detection result to update, and calculate the target position estimate with the minimum mean square error as the final position to update the trajectory. This not only completes the update of the trajectory state, but also

corrects the target position. However, there are no detections when the trajectory extension is updated, so only the predicted position is updated to the trajectory.

---

**Algorithm 1:** Pseudo-code of FCT.

---

**Input:** A video sequence $V$; object detector with embeddings Det; Kalman Filter KF; tracking thresholds $T_{emb}$, $T_{iou}$ and $T_{init}$; trajectory expansion threshold $T_{extd}$; confidence decay factor $\alpha$
**Output:** Tracks $T$ of the video

1.　　$T \leftarrow \varnothing$ (Initialization)
2.　**for** $f$ in $V$ **do**
3.　　　$D \leftarrow \text{Det}(f)$ / * predict detection boxes, confidences and embeddings * /
4.　　　$D_{emb} \leftarrow$ Targets in $D$ with greater confidence than $T_{emb}$
5.　　　$D_{iou} \leftarrow$ Targets in $D$ with confidence between $T_{iou}$ and $T_{emb}$
6.　　　**for** $t$ in $T$ **do** / * predict new locations of tracks * /
7.　　　　　$t \leftarrow \text{KF}(t)$
8.　　　**end**

　　　/ * embedding matching * /

9　　　Associate $T$ and $D_{emb}$ using embedding feature similarity
10　　$D_{remain} \leftarrow$ remaining object boxes from $D_{emb}$
11　　$T_{remain} \leftarrow$ remaining tracks from $T$

　　　/ * IoU matching * /

12　　$D_{iou} \leftarrow D_{iou} \cup D_{remain}$
13　　Associate $T_{remain}$ and $D_{iou}$ using IoU distance and update $T$
14　　$D_{unuse} \leftarrow$ remaining object boxes from $D_{iou}$
15　　$T_{lost} \leftarrow$ remaining tracks from $T_{remain}$
16　　**for** $t$ in $T_{lost}$ **do**
17　　　　**if** $t.score > T_{extd}$ & $t.len > 2$ & $t.activated$ **then** / * trajectory expansion * /
18　　　　　　$t.score = t.score \times \alpha$
19　　　　　　update $T$ with $t$
20　　　　**else**
21　　　　　　$T \leftarrow T \backslash \{t\}$ / * delete unmatched tracks * /
22　　　　**end**
23　　**end**
24　　**for** $d$ in $D_{unuse}$ **do**
25　　　　**if** $d.score > T_{init}$ **then**
26　　　　　　$T \leftarrow T \cup \{d\}$ / * initialize new tracks * /
27　　　　**end**
28　　**end**
29　**end**
30　**return** $T$

---

If the track is not judged to be a missing track, we will mark it as a lost track. Tracks marked as lost are still retained in $T$ and participate in subsequent matching (but not output as tracks in the current frame), and they will not be deleted until the loss time reaches a certain range. For the unmatched target $D_{unuse}$, if the target confidence is greater than the trajectory initialization threshold $T_{init}$, the target will be initialized as a new trajectory; otherwise, it will be discarded. Finally, the tracking trajectory of the video is returned.

## 3. FTC Tracker

We use YOLOX-s to build a one-shot multi-target tracker for various experiments. The model structure of YOLOX-s is shown in Figure 2.

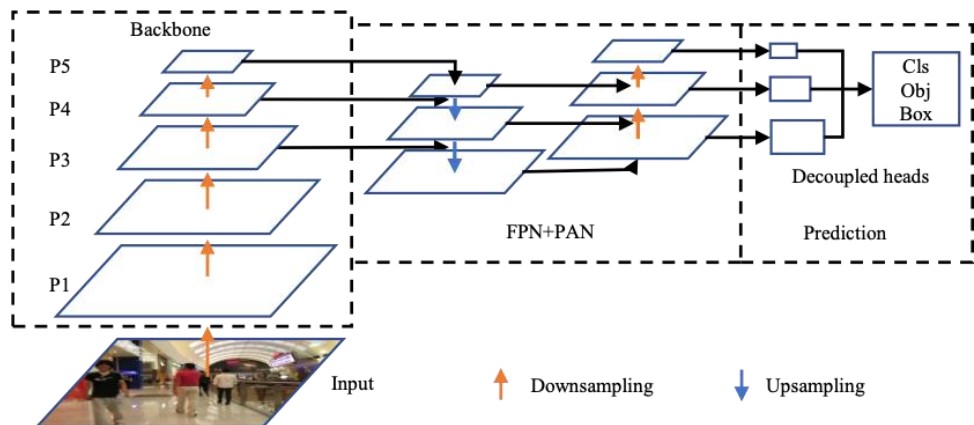

**Figure 2.** YOLOX-s.

The input part uses Mosaic [4] and Mixup [13] for data enhancement. The Mosaic method randomly stitches four images in random scaling, random cropping and random arrangement, which not only enriches the data but also adds many small objects to the dataset, improving the robustness of the model to small objects. Mixup was originally derived from the image classification task. It realizes data expansion by filling and scaling different images to the same size and then performing weighted fusion. This can steadily improve the classification accuracy with almost no computational overhead. The backbone network uses a cross-stage partial network, and the neck part uses feature pyramid networks and path aggregation networks for feature fusion to enhance the detection ability.

YOLOX improves the prediction module of YOLO into an anchor-free-based decoupling head and uses effective label assignment strategy SimOTA [14]. One of the embedded decoupled heads is shown in Figure 3. We embed the appearance model (Emb) into the classification decoupling head with minimal computational cost.

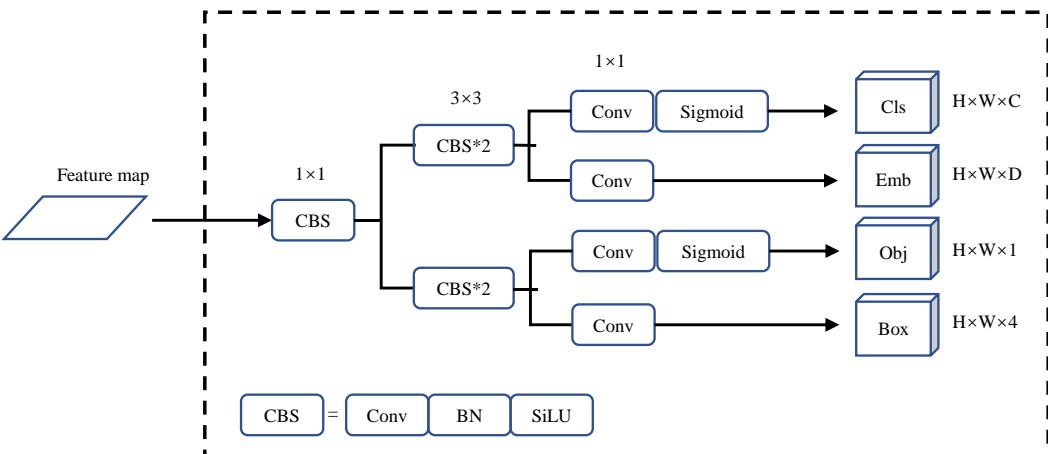

**Figure 3.** Embedded structure.

The Emb branch needs to output a $D$-dimensional vector for each prediction target. Therefore, the output feature map size is $H \times W \times D$. The choice of $D$ value is very important. If the dimension is too low, the target features cannot be accurately expressed, and if the dimension is too high, it will not only affect the training and tracking speed, but also will bring about an imbalance between the dimensions of detection and re-identification (Re-ID) features. Therefore, we compared different dimension values, such as 64, 128, 256 and 512, and finally selected the feature dimension as 128.

For the selection of positive samples of the prediction frame, YOLOX adopts preliminary screening and the SimOTA algorithm. There are two types of preliminary screening:

one is to judge according to the range of the label box and the other is to judge according to the center point of the label box. However, the positive samples are not completely suitable for Re-ID training, because, although some sample prediction boxes are within the range of the screening conditions, their center points are far away from the ground truth (GT) center. This makes the background part in the prediction frame excessive, exceeding the foreground part, which easily leads to blurred Re-ID features. This is not conducive to the re-identification task. In Figure 4, the dotted boxes represent squares of different scales, the black box is the target GT position, and the yellow box is the predicted position. Points of different colors are the center points of the corresponding target. The background part in the yellow prediction box is redundant with the foreground part.

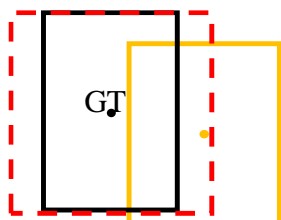

**Figure 4.** Screening process of detection positive samples.

To reduce the feature blur problem, we propose a secondary screening strategy for embedding positive samples. By comparing the distance between the center of the predicted positive sample and the center of the GT, the predicted positive samples are screened for a second time. More specifically, first, the offset range is calculated according to the GT detection frame. A rectangle with a long side of 5 is set on the three feature maps according to the height–width ratio of the labeling frame, and multiplied by the corresponding downsampling multiple to obtain its rectangular range on the input image. In order to ensure that the center point of the predicted positive sample is within the range of the label, the minimum edge length of the above rectangle and the label box is taken as the final offset range. Then, the offset of the current predicted location in the x-axis and y-axis directions for screening is obtained, and the embedding positive sample is achieved after secondary screening.

Figure 5 shows the process of the secondary screening, where the dotted box is the offset range, the black box is the target position, the green box is the predicted position beyond the range, and the yellow box is the predicted position within the range. Points of different colors are the center points of the corresponding target. It can be seen from the rightmost group that when the offset range exceeds the callout frame, the callout frame range shall prevail.

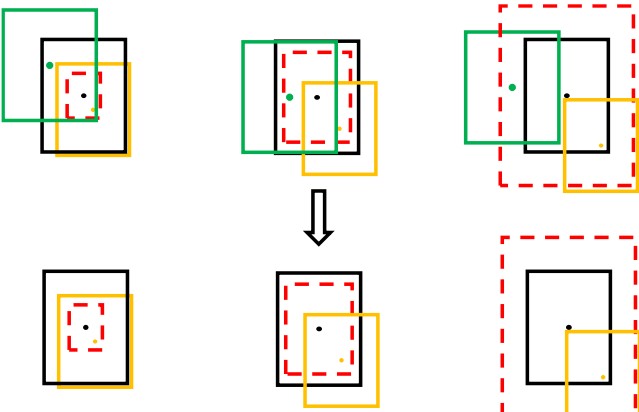

**Figure 5.** Secondary screening.

We treat embedding appearance re-identification training as a classification task [15]. Objects with the same ID in the dataset are regarded as the same category. During the training process, the embedding feature vector of the target is sent to a linear classification layer to obtain the probability value of each category. We use cross-entropy loss for ID classification training. The classifier does not need to be used during testing; only the embedding features are subjected to the association process to calculate the similarity. When testing, the classifier does not need to be used; we only need to submit the embedding features to the association process to calculate the similarity. The category loss (Cls branch) and object confidence loss (Obj branch) use binary cross-entropy to calculate the loss. Box loss uses IoU loss calculation only when Mosaic enhancement is used, and it increases $L_1$ loss when Mosaic and Mixup enhancement is turned off.

## 4. Experiments

### 4.1. Experimental Settings

In this section, we use the MOT17 [16] and MOT20 [17] datasets for experimental validation. Specifically, for the ablation experiments, we split the MOT17 training set into train_half and val_half for training and validation, respectively. When comparing with different algorithms, we employed the complete training dataset, and submitted the test data to the MOT Challenge official website to obtain the evaluation results. When using the MOT17 dataset, we set the input image size to be $608 \times 1088$. Since the MOT20 dataset is denser, for the purpose of obtaining a better tracking effect, the input image was increased to $896 \times 1600$.

As for the evaluation metrics, we adopted the MOT Challenge Benchmark, including Multiple Object Tracking Accuracy (MOTA), Identification F1-Score (IDF1), False Positive (FP), False Negative (FN), ID switch (IDs), Mostly Tracked trajectories (MT) and Mostly Lost trajectories (ML). MOTA is the tracking accuracy rate, which is calculated based on FPs, FNs, and IDs. IDF1 is the ratio of correctly identified detections over the average number of ground-truth and computed detections. MT is the ratio of ground-truth trajectories that are covered by a track hypothesis for at least 80% of their respective life span. ML is the ratio of ground-truth trajectories that are covered by a track hypothesis for at most 20% of their respective life span. Meanwhile, we also used Frames Per Second (FPS) to evaluate the running speed. In all experimental results, ↑ means that the larger the better, and ↓ means that the lower the better.

The experimental hardware environment was a deep learning server with an Intel Xeon CPU E5-2650 v4, 2.2 GHz processor and Tesla K80 graphics card (4 photos). The algorithm model was initialized by using the model parameters of YOLOX-s during training. The initial learning rate was set to $10^{-3}$, and we used 1 epoch warm-up and cosine annealing schedule [18] for training. The optimizer was stochastic gradient descent (SGD) with weight decay of $5 \times 10^{-4}$ and momentum of 0.9. We trained the model for 50 epochs with batch size 12, and turned off Mixup and Mosaic at the 40th epoch. The threshold settings involved in Algorithm 1 are shown in Table 1.

**Table 1.** Threshold settings.

| Thresholds | MOT17 | MOT20 |
|:---:|:---:|:---:|
| $T_{emb}$ | 0.60 | 0.60 |
| $T_{iou}$ | 0.30 | 0.30 |
| $T_{init}$ | 0.70 | 0.70 |
| $T_{extd}$ | 0.75 | 0.35 |
| $\alpha$ | 0.85 | 0.75 |

### 4.2. Ablation Studies

First, the selection of the embedding feature dimension was experimentally analyzed. In order to choose the appropriate dimension, we chose the dimension values of 64, 128, 256 and 512 for comparison, as shown in Table 2. Although increasing the feature dimension

can reduce FN, it can also increase FP, and the tracking effect does not improve with the increase in the feature dimension. On the contrary, according to the results in Table 2, when the lowest dimension is 64, there is a higher IDF1, which shows that the lower feature dimension is more suitable for the model in this work. When the dimension value is 128, it has the highest MOTA and the highest IDF1 value, and the overall tracking effect is more balanced.

**Table 2.** Tracking effect of different feature dimensions.

| Dim | MOTA↑ | IDF1↑ | FP↓ | FN↓ | IDs↓ | FPS↑ |
|---|---|---|---|---|---|---|
| 64 | 68.4 | 69.6 | **1308** | 15,324 | 374 | **25.1** |
| 128 | **68.8** | **69.9** | 1413 | 15,050 | **348** | 24.7 |
| 256 | **68.8** | 66.9 | 1553 | 14,790 | 491 | 24.4 |
| 512 | 68.5 | 64.9 | 1710 | **14,703** | 543 | 23.5 |

Table 3 shows the comparison of model parameters and calculations in different dimensions. We can see that the embedding of the appearance model only increases the parameters and calculations by a very small amount. Therefore, in order to achieve a better tracking effect, we set the Re-ID embedding features in these experiments to be all 128-dimensional.

**Table 3.** Parameters and calculations in different dimensions.

| Dim. | Params (M)↓ | Gflops↓ |
|---|---|---|
| base | 8.94 | 43.02 |
| 64 | 8.96 | 43.24 |
| 128 | 8.99 | 43.46 |
| 256 | 9.04 | 43.91 |
| 512 | 9.14 | 44.81 |

Next, ablation experiments were performed on the different improved strategies, as shown in Table 4. It was predicted that the secondary screening of positive samples could significantly reduce the number of missed follow-ups. The MOTA was increased by 1.1%, but the false follow-up and ID switching of the target were increased, and the IDF1 was decreased by 0.6%. On this basis, adding the tracking threshold separation strategy can effectively improve the model tracking effect. Although adding low-confidence targets in the secondary matching produces more misjudgments and leads to an increase in FP, it can reduce the loss of targets, the FN value is greatly reduced, and the MOTA is increased by 0.8%. IDF1 was improved by 0.3%. After restricting the update of low-confidence target embedding features (H-conf), IDF1 increased by 0.3%, and IDs also decreased, indicating that reducing low-quality features in the trajectory is indeed conducive to strengthening the feature matching effect of the algorithm, and the target trajectory is more stable. However, the lack of the latest features of the target can also bring certain tracking of errors and tracking of omissions, which have a certain impact on the tracking accuracy, and the MOTA is reduced by 0.1%.

**Table 4.** Ablation experiments with different improvements.

| Methods | MOTA↑ | IDF1↑ | FP↓ | FN↓ | IDs↓ |
|---|---|---|---|---|---|
| Baseline | 68.8 | 69.9 | **1413** | 15,050 | **348** |
| S-screening | 69.9 | 69.3 | 1812 | 14,019 | 366 |
| T-separation | 70.7 | 69.6 | 2503 | 12,913 | 376 |
| H-conf | 70.6 | 69.9 | 2537 | 12,927 | 365 |
| FTC | **70.9** | **70.8** | 2568 | **12,790** | **348** |

Track compensation uses the predicted track position to expand the missing track, which can reduce the information loss and fragmentation of trajectories. It can be seen

from Table 4 that the experimental results are consistent with the analysis. Due to the supplementation of the track compensation to the missing track, the FN decreases by 137, and the reduction in the track fragmentation increases the IDF1 by 0.9%. This increases the number of traces that can hold the target ID for a long time and reduces the IDs by 26. At the same time, due to the misjudgment of the normal termination trajectory and the inaccuracy of the predicted position, the number of false calls increased, and the FP increased by 31, but the number of missed calls that were successfully supplemented was greater than the number of false calls added, so the tracking accuracy was also improved to a certain extent. MOTA increased by 0.3%. These improvements only add some judgment and processing procedures in the association stage, and the increase in the amount of calculation is very small, so it will not affect the real-time tracking.

*4.3. Robustness Experiment*

In the tracking threshold separation strategy, the selection of the tracking threshold for secondary matching is also very important. To explore the impact of different thresholds on tracking performance, we took the IoU tracking threshold at intervals of 0.1 between 0.1 and 0.6, and checked the tracking effect of different thresholds, as shown in Figure 6. The threshold value of 0.6 is the baseline effect of the strategy.

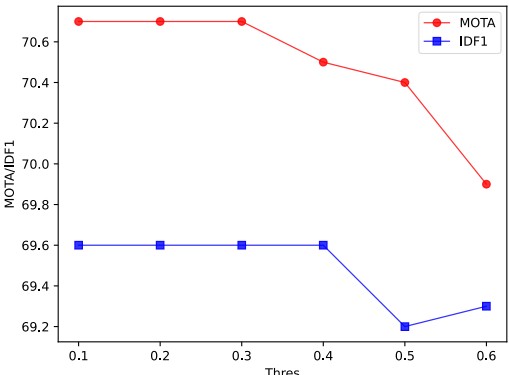

**Figure 6.** $T_{iou}$ fluctuation experiment.

Figure 6 shows that as the IoU tracking threshold decreases, the tracking accuracy gradually increases, but when the threshold is reduced to 0.4 and 0.3, IDF1 and MOTA no longer change. This is because, when performing IoU matching, the calculation of the matching cost needs to be combined with the target confidence. The lower the confidence, the higher the matching cost. Therefore, when the target confidence is lower than a certain range, it no longer affects the tracking effect. Although setting the IoU tracking threshold to a minimum value of 0.1 can ensure that the model has better tracking performance, targets with a confidence level between 0.1 and 0.3 still participate in the association process, resulting in unnecessary calculations. Therefore, the IoU tracking threshold can be taken as the maximum value of 0.3 to stabilize the model tracking performance.

In order to fully demonstrate the improvement of the tracking effect by the tracking threshold separation strategy, a visual analysis was performed on different video sequences of the MOT17 val_half data, and the same adjacent frames before and after the improvement were compared. To ensure the comparison effect, we only printed the information of the comparison target, as shown in Figure 7. ID is the target ID number, and S is the detection confidence.

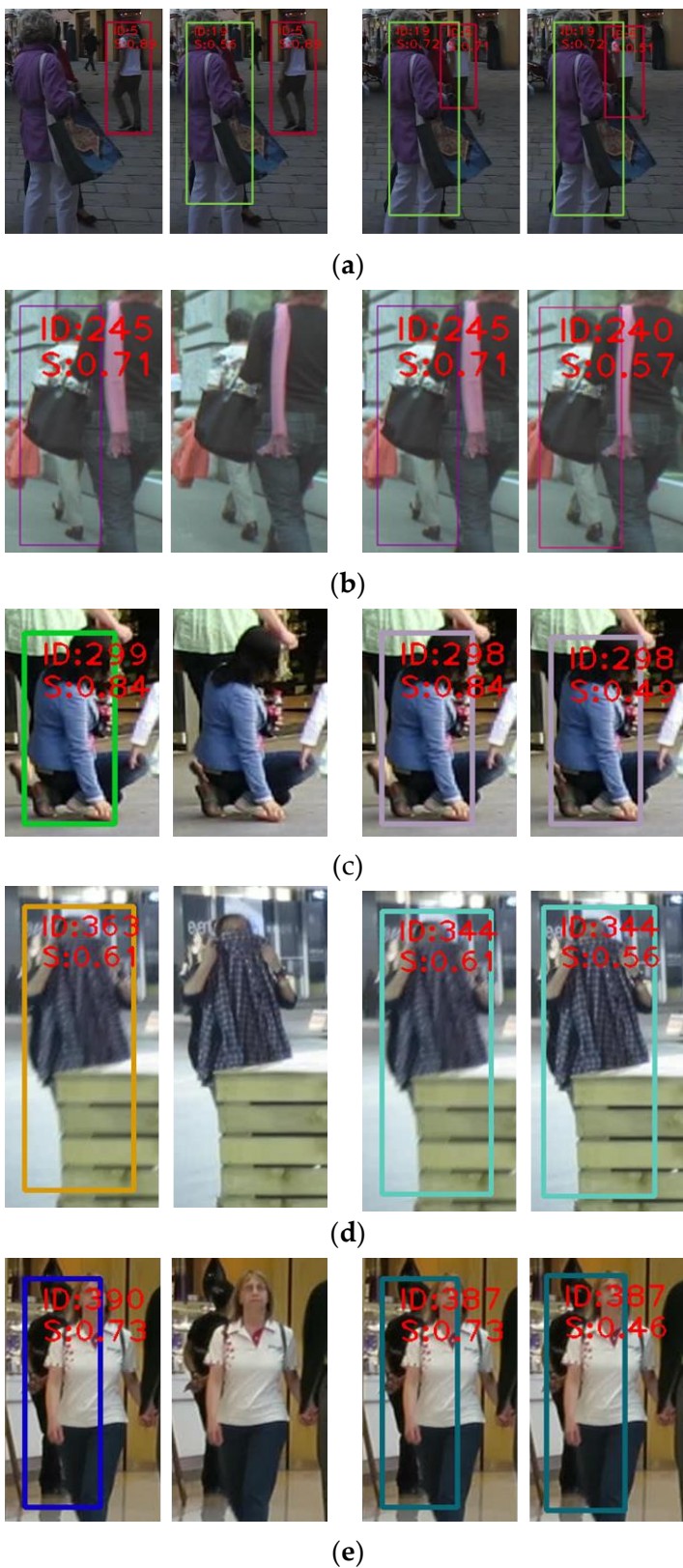

**Figure 7.** *Cont.*

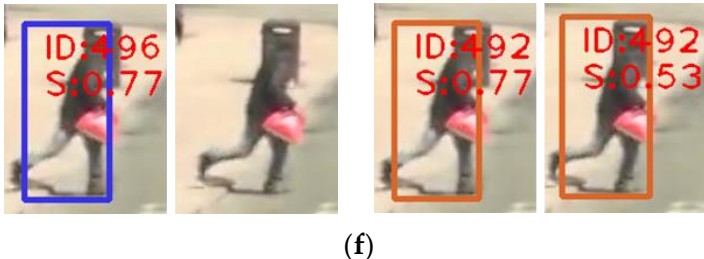

(**f**)

**Figure 7.** Tracking effect of tracking threshold separation strategy. (**a**) MOT17-02 frame 16 and 26. (**b**) MOT17-05 frame 57 and 58. (**c**) MOT17-09 frame 36 and 37. (**d**) MOT17-10 frame 201 and 202. (**e**) MOT17-11 frame 17 and 18. (**f**) MOT17-13 frame 295 and 296.

Figure 7a shows two sets of comparison targets. The two images on the left are the comparison effects of the 16th frame before and after the improvement. The target with ID number 19 is severely occluded by the target in front of it. Before the improvement, the target was missed. After the improvement, the predicted box with a confidence value of 0.56 was associated to the trajectory. The two images on the right show the comparison effect of the 26th frame. Although the target with ID number 5 is not missed, the detection frame with a confidence level of 0.71 on the left is obviously not as close to the target as the detection frame with a confidence level of 0.51 on the right. In Figure 7b, the two images on the left are the 57th and 58th frame tracking images before the improvement, and the two groups on the right are the tracking effect images of the two frames after the improvement. After the improvement, The low-confidence detection is successful linked to trajectory. The remaining images from Figure 7c–f are the same as in Figure 7b, showing the comparison of the tracking effects of two adjacent frames. Due to the different tracking results before and after the improvement, some targets have different IDs in different results, as shown in Figure 7c, but they all refer to the same target in the same frame.

From the results of the ablation experiments, we can see that trajectory compensation is a judgment and supplementation to the tracking of the trajectory. Although reducing the leakage of tracking can improve the stability of the trajectory, misjudgment can also increase the false tracking of the trajectory and reduce the tracking accuracy. To verify the effectiveness of the missing trajectory compensation in the trajectory compensation algorithm and explore its advantages and disadvantages under different video conditions, the changes in FP and TP in different video sequences of the dataset were analyzed, and the results are shown in Figure 8. The abscissa represents different video sequences, and the ordinate represents the numerical variation. For fairness, uniform thresholds were employed for different video sequences.

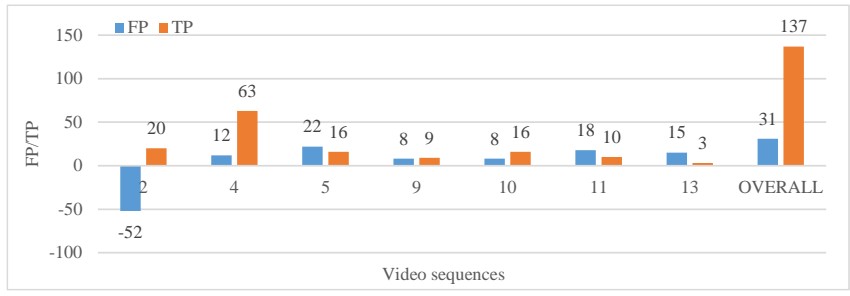

**Figure 8.** FP and TP changes.

Figure 8 shows that the number of newly added TPs in 02, 04, 09 and 10 is higher than the number of newly added FPs. Among them, the 02 and 04 sequences have the best effect, and the FP in the 02 sequence greatly decreases. From the overall data, the newly added TP is much higher than the newly added FP, but the effect of the 05, 11 and 13 sequences is poor. Although it can be improved by adjusting the parameters individually, uniform



parameters are used considering the overall nature of the dataset. In order to investigate the reasons for the difference in the effects of different video sequences, we analyzed the video data and found that there was camera motion in the video sequences 05, 10, 11 and 13, while the rest were captured by fixed cameras. At the same time, the range of the 05 and 11 sequences of the video crowd is small, and there are many large-scale occlusions. The camera motion and occlusion can affect the prediction effect of the trajectory, so the 05 and 11 sequences are less effective. The 13 sequences were taken by in-vehicle equipment. Although the field of view is very wide, the target scale is small and there is a wide range of camera motion due to vehicle driving and steering. This renders the Kalman filter unable to accurately estimate the trajectory position. Therefore, trajectory compensation is more suitable for trajectories with more accurate motion information estimation.

When setting the judgment conditions for missing trajectories, the selection of the trajectory expansion threshold $T_{extd}$ and the confidence attenuation coefficient $\alpha$ is also very important, where $T_{extd}$ uses the last matching target confidence of the trajectory to filter the missing trajectories, and controls the expansion times of the missing trajectories. According to experimental experience, $T_{extd}$ should be greater than or equal to the tracking threshold, and it should be selected between the tracking threshold and 1. $\alpha$ should be selected between $T_{extd}$ and 1.

In order to explore the change in tracking effect when different parameter values are set, the tracking effect with different $T_{extd}$ and $\alpha$ was experimentally analyzed. The tracking threshold $T_{emb}$ is 0.6, so the value range of $T_{extd}$ and $\alpha$ is 0.6 to 1, and the interval is set to 0.05, as shown in Figure 9.

Since it is meaningless when $T_{extd}$ and $\alpha$ are set to 1, and the result changes significantly when $\alpha$ is set to 0.9, only six ranges of experimental results are presented. The two dotted lines represent their baselines, respectively. We can see that when different parameters are taken, the two indicators of MOTA and IDF1 are basically higher than the baseline, and the tracking effect tends to improve gradually as the two parameters decrease, indicating that the reduction relaxes the judgment conditions for missing tracks, and the supplementation for missing tracks increases.

However, it is not true that the more the trajectory is expanded, the better the tracking effect will be. As shown in Figure 9e,f, after $T_{extd}$ drops to a certain extent, the two indicators both decrease, and as $\alpha$ decreases, the two indicators gradually increase. This means that the expansion times of some low-scoring trajectories are too high, which can cause many false follow-ups and affect the tracking effect. It can be improved by appropriately reducing the number of expansions through $\alpha$. Therefore, the selection of the two parameters should be moderate, and it is not easy to take too large or too small values.

### 4.4. Applications on Other Algorithms

Although the threshold separation strategy has certain limitations, the FTC algorithm has good scalability. In order to fully verify the scalability of FTC, we applied FTC to several different correlation algorithms under the same detection model and different correlation algorithms. The FTC was also extended to several mainstream multi-target tracking algorithms to verify its scalability under different types of algorithms.

Firstly, the expansion experiments of different correlation algorithms were carried out. With YOLOX-s as the detector, four different correlation algorithms, Sort [19], DeepSort [20], MOTDT [10] and JDE [21], were selected. The detector was trained on MOT17 train_half, and the tracking effect was tested on val_half. The results are shown in Table 5. After using FTC for trajectory compensation, both MOTA and IDF1 have a certain improvement, and the amount of ID switching is also reduced. This shows that the problem of missing information is common in different association algorithms.

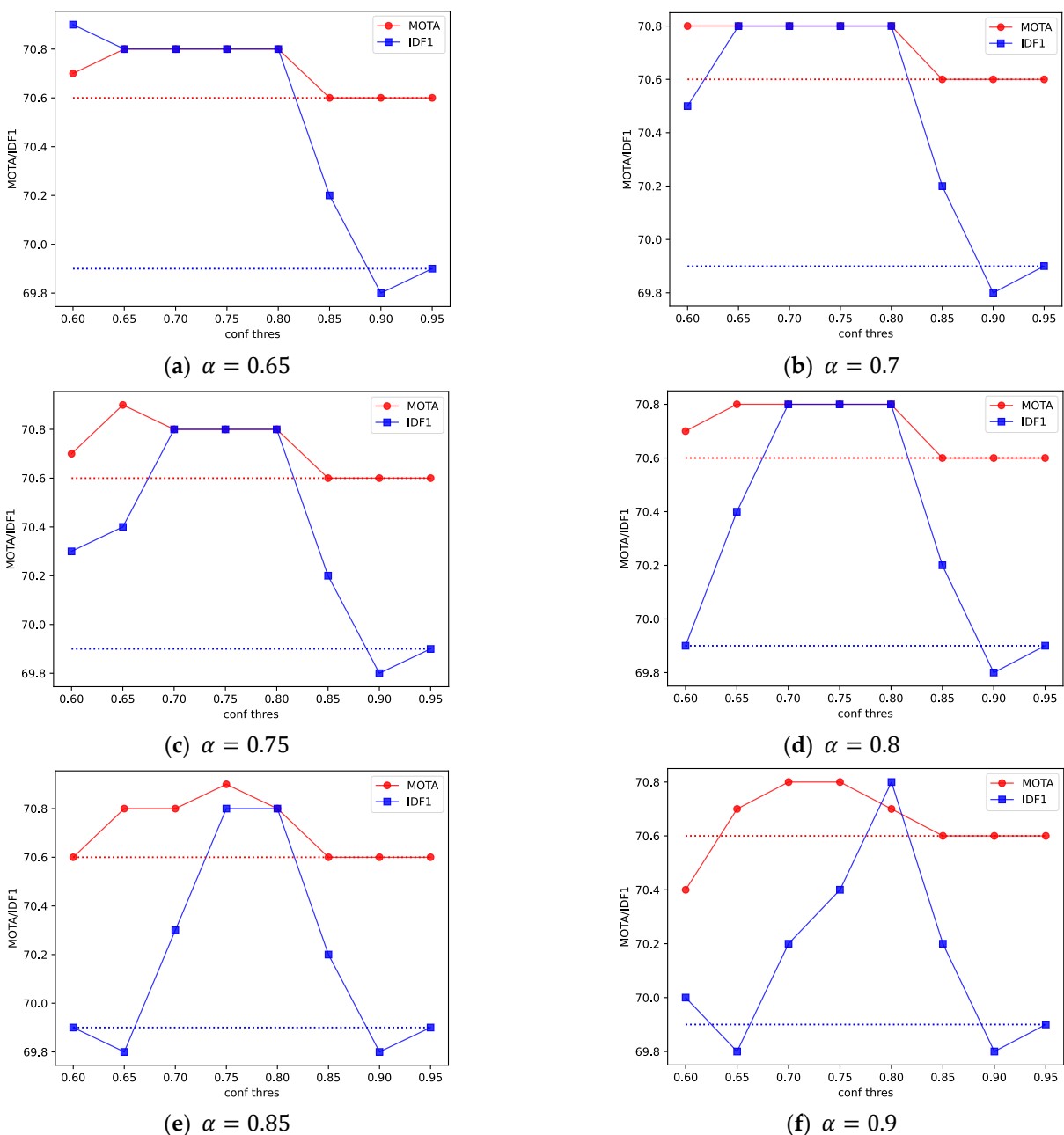

**Figure 9.** Comparison of different $T_{extd}$ and $\alpha$.

**Table 5.** Extended experiments on different association algorithms.

| Algorithm | FTC | MOTA↑ | IDF1↑ | FP↓ | FN↓ | IDs↓ |
|---|---|---|---|---|---|---|
| Sort [19] |  | 67.6 | 69.2 | 3309 | 13,590 | 566 |
|  | √ | 68.1(+**0.5**) | 69.3(+**0.1**) | 3576 | 13,043 | 551 |
| DeepSort [20] |  | 69.4 | 71.5 | 2486 | 13,765 | 250 |
|  | √ | 69.6(+**0.2**) | 72.7(+**1.2**) | 3262 | 12,889 | 241 |
| MOTDT [10] |  | 69.2 | 71.2 | 2765 | 13,426 | 412 |
|  | √ | 69.6(+**0.4**) | 72(+**0.8**) | 3199 | 12,841 | 364 |
| JDE [21] |  | 69.9 | 69.3 | 1812 | 14,019 | 366 |
|  | √ | 70.4(+**0.5**) | 70.3(+**1.0**) | 2558 | 13,061 | 319 |

It shows that FTC can judge and compensate for the missing trajectories in different tracking paradigms. Meanwhile, the increase in FP and decrease in FN in the experimen-

tal results of the different correlation algorithms are consistent with the analysis of the compensation principle of FTC in this paper, and its accuracy is verified again. Observing the changes in MOTA and IDF1 for different algorithms, we found that the improvement in IDF1 of the other algorithms except Sort is higher than that of MOTA. This is because the addition of missing heels can reduce the problem of trajectory fragmentation. Therefore, although FTC supplements fewer leaks, it can better solve the problem of trajectory fragmentation. However, the Sort algorithm does not use feature similarity matching, and the lost track retrieval ability is poor, so the improvement in IDF1 is lower compared to other algorithms. MOTDT uses the trajectory prediction position to supplement the detection candidate frame before performing matching and association, but after applying FTC, MOTA increases by 0.4%, and IDF1 increases by 0.8%, indicating that supplementing detection information cannot fully solve the problem of missing trajectories.

In addition, extended experiments were also carried out on different mainstream multi-target tracking algorithms, and four different algorithms, JDE [21], FairMOT [15], CenterTrack [22] and CTracker [23], were selected. In Table 6, the letter K indicates that the algorithm uses the Kalman filter to predict the trajectory. Among them, JDE and FairMOT are both one-shot algorithms based on the JDE correlation algorithm. The difference is that JDE uses YOLOv3 [3] as the detector and FairMOT uses CenterNet [1] as the detector. Table 6 shows that FTC still achieves a good improvement effect under different detection models.

**Table 6.** Extended experiments on different tracking algorithms.( $\sqrt{}$ indicates that the method here is used).

| Algorithm | FTC | MOTA↑ | IDF1↑ | FP↓ | FN↓ | IDs↓ |
|---|---|---|---|---|---|---|
| JDE(K) [21] | | 74.3 | 69.1 | 5236 | 22,319 | 1343 |
| | $\sqrt{}$ | 74.5(+**0.2**) | 70.9(+**1.8**) | 6448 | 21,000 | 1169 |
| FairMOT(K) [15] | | 83.8 | 81.9 | 2712 | 14,877 | 553 |
| | $\sqrt{}$ | 84.6(+**0.8**) | 82.8(+**0.9**) | 4349 | 12,429 | 461 |
| CenterTrack [22] | | 70.9 | 65 | 2602 | 28,853 | 1246 |
| | $\sqrt{}$ | 70.6(−**0.3**) | 65.8(+**0.8**) | 3590 | 28,358 | 1111 |
| CTracker [23] | | 76.4 | 67.8 | 1783 | 23,791 | 976 |
| | $\sqrt{}$ | 76.4(+**0.0**) | 69.3(+**1.5**) | 1786 | 23,794 | 941 |

Since the algorithms in the current experiments all used the Kalman filter to predict the trajectory position, CenterTrack and CTracker were chosen to verify the scalability of FTC without the Kalman filter. Both algorithms use adjacent image pairs as input. The difference is that CenterTrack estimates the target position by outputting the center point offset of the previous frame of the target in the current frame through the deep network, while CTracker calculates the target offset distance through a simple velocity model. However, the center point offset output by CenterTrack is bound to the predicted candidate frame, and CTracker outputs the predicted candidate frame in the form of detection pairs, so it can only predict the trajectory position of adjacent frames, and cannot perform long-term expansion of missing trajectories. Therefore, only the trajectory expansion threshold is set when applying FTC to CenterTrack and CTracker, and only one trajectory expansion is performed for missing trajectories.

The experimental results show that although there is no Kalman filter for the long-term prediction of missing trajectories, it still achieves a good improvement effect. The IDF1 of CenterTrack has an improvement of 0.8%, and the IDF1 of CTracker has an improvement of 1.5%, once again verifying the FTC scalability advantage. However, MOTA has not been improved, indicating that the trajectory position prediction is not as accurate as the Kalman filter.

*4.5. MOT Challenge Result*

In this section, the MOT Challenge test set is compared with different algorithms, and all the test results were obtained from the MOT Challenge official website. Since the test

set annotation is not public, and the datasets used for each algorithm test are different, we used the MOT17 and MOT20 test sets to compare different algorithms.

Table 7 shows the results on the MOT17 test set. Our algorithm achieves higher tracking accuracy and has a much faster online tracking speed than other algorithms. Table 8 shows the comparison results of different algorithms on the MOT20 test set. Due to the denser pedestrians, there are more crowded scenes and occlusions in the MOT20 dataset. Therefore, we set the input image size to $896 \times 1600$ to achieve a better tracking effect. However, the FPS drops by approximately 2.5 at the expense of some tracking speed. The increase in the number of targets in the data also increases the amount of tracking computation, so the tracking speed in the MOT20 dataset decreases.

**Table 7.** Comparison of different tracking algorithms on MOT17.

| Algorithm | MOTA↑ | IDF1↑ | MT↑ | ML↓ | IDs↓ | FPS↑ |
|---|---|---|---|---|---|---|
| SST [24] | 52.4 | 49.5 | 21.4 | 30.7 | 8431 | <1.0 |
| TubeTK [25] | 63.0 | 58.6 | 31.2 | **19.9** | 4137 | <1.0 |
| CTracker [23] | 66.6 | 57.4 | 32.2 | 24.2 | 5529 | 1.7 |
| CenterTrack [22] | 67.3 | 59.9 | 34.9 | 24.8 | 2898 | 6.0 |
| FairMOT [15] | 69.8 | **69.9** | 38.2 | 21.0 | 3996 | 7.0 |
| TransCenter [26] | 70.0 | 62.1 | 38.9 | 20.4 | 4647 | <1.0 |
| FTC (Ours) | **70.8** | 69.2 | **41.4** | 22.9 | **2526** | **24.4** |

**Table 8.** Comparison of different tracking algorithms on MOT20.

| Algorithm | MOTA↑ | IDF1↑ | MT↑ | ML↓ | IDs↓ | FPS↑ |
|---|---|---|---|---|---|---|
| MLT [27] | 48.9 | 54.6 | 30.9 | 22.1 | **2187** | 1.0 |
| FairMOT [15] | 61.8 | **67.3** | **68.8** | **7.6** | 5243 | 5.0 |
| TransCenter [26] | 61.9 | 50.4 | 49.4 | 15.5 | 4653 | <1.0 |
| FTC (Ours) | 65.0 | 65.3 | 61.4 | 10.4 | 5312 | **8.5** |

Through the sufficient training of a large amount of MOT20 data and large-scale image input, our algorithm has 65.0% MOTA and 65.3% IDF1, while still maintaining the fastest tracking speed of 8.5 FPS. Compared with the test results of MOT17, although the number of targets in the MOT20 dataset is larger, each algorithm can obtain a higher trajectory hit rate ML and a lower trajectory loss rate ML. This should be related to the fact that the MOT20 data are captured at high places. This means that the occlusion range between targets is small and there is no large-scale crowd occlusion. Thus, our algorithm also has higher MT and lower ML, but there is more ID switching.

## 5. Conclusions

We proposed a simple and effective data association algorithm. The algorithm realized the reasonable distribution of detection results through the tracking threshold separation strategy, and then used the trajectory prediction information to compensate for the missing target information, so as to fully utilize the detection and tracking information. After comparing it with mainstream correlation algorithms such as Sort, DeepSort and JDE, as well as MOTDT supplementing detection, the effectiveness of our method is verified. Furthermore, the process adds only a small amount of computation to data association and has little impact on real-time performance. We also proposed an efficient one-shot tracker, namely FTC Tracker, which achieved high tracking accuracy and high online tracking speed. Compared with mainstream algorithms such as FairMOT and TransCenter, FTC Tracker has great advantages in tracking accuracy and speed. In addition, FTC not only has a stable improvement effect, but also has strong scalability, and can be widely used in various association algorithms or multi-target tracking algorithms. FTC is simple and easy to use, but, due to the dependence on parameter settings, the improvement effect on different methods is not stable enough. In the future, we will consider how to implement FTC in

an adaptive way, eliminating the need for parameter adjustment work and improving its stability.

**Author Contributions:** All authors contributed to the study conception and design. Conceptualization, methodology, investigation, formal analysis, writing—original draft preparation, J.J., L.W. and Q.Y.; writing—review and editing, supervision, J.S. All authors have read and agreed to the published version of the manuscript.

**Funding:** This research was funded in part by the National Key Research and Development Program of China (grant no: 2018YFC1603303, 2018YFC1604004) and the National Science Foundation of China (grant no: 61672263).

**Institutional Review Board Statement:** Not applicable.

**Informed Consent Statement:** Not applicable.

**Data Availability Statement:** Not applicable.

**Conflicts of Interest:** The authors declare no conflict of interest.

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
