# Peer review of "Multi-Object Tracking Algorithm of Fusing Trajectory Compensation"

_mathematics, doi:10.3390/math10152606_

Round 1

Reviewer 1 Report

This paper takes up the challenging problem of multiple objects tracking by proposing a new algorithm based on fusing trajectory compensation. The performance of this approach is illustrated through experiments on many multiple objects tracking datasets. Moreover, to validate the robustness of their proposed algorithm, the authors applied it to different association algorithms and tracking algorithms. The algorithm is well-described and the experimental results fully support the authors’ approach. In addition to the English of the paper, the main concern that I have is that I believe is not suitable for a mathematics journal. The paper does include any mathematical or algorithmic foundations; and therefore, I think it would be more convenient to submit it to other MDPI journal like Journal of Imaging for instance. In case the authors decide to submit elsewhere, I am including few comments and issues I have detected during the reading of the manuscript.

1.     All abbreviations should be defined before first use. Those in the abstract should be defined in the same place to make it self-contained, and then redefined if needed in the body of the paper (e.g., IoU).

2.      Page 1: Add references to the approaches presented in the last paragraph. In particular, at least one reference for each approach (end-to-end trajectory modeling, supplement of detection information).

3.      Line 73: Change “and assuming” to “and assume”.

4.     The authors should pay attention to the use of indefinite and definite articles in the whole text. E.g., in line 77, “is trajectory” should be “is a trajectory”; in line 78, “is confidence” should be “a is confidence”; in line 79, “identity of object” should be “the identity of the object”, and so on. Actually, there are plenty of cases.

5.      Algorithm 1: Change “tracking threshold” to “tracking thresholds”.

6.      Line 210: The image size is missing.

7.    Line 260: Rewrite the sentence “Increased, reduced ID switching, and reduced IDs by 17.”

Reviewer 2 Report

Manuscript entitled 'Multi-Object Tracking Algorithm of Fusing Trajectory Compensation' is an interesting paper that proposes a multi-object tracking algorithm of fusing trajectory compensation. In my opinion it was written correctly, the research is convincing and the literature review is adequate. I only have a few minor comments:

1. I think in the introduction you should describe the structure of the article - what is included in the individual sections.

2. Are you sure that the title of Algorithm 1 should contain word 'CT'?

3. You should briefly describe what the abbreviations of the indicators used in the research (eg IDF1) mean.

4. Planned further work should be included in the conclusions. Additionally, a discussion with references to literature would also be useful in this section.

5. There are some linguistic mistakes in the paper (e.g. line 178 'input image. The rectangular range', references to literature [14] and [17] are written in superscript, there is no spaces in line 247 ('features (H-conf)' ) and 453 ('8.5FPS')) - please check the manuscript.

Reviewer 3 Report

Comments and Suggestions for Authors:  

In this paper, the author has proposed a simple and effective data association algorithm for multi-object tracking algorithm. The algorithm realized the reasonable distribution of detection results through the tracking threshold separation strategy, and then used the trajectory prediction information to compensate for the missing target information, so as to fully utilize the detection and tracking information.

However, I have some suggestions as follows:

1.      The overall writing is ok.

2.      However, the method is needed to improve with more clarity!

3.      There are mistakes at line no. 209, correct the line no. 209 at page 7.

4.      There are mistakes at line no. 255-256, correct the line no. 255-256 at page 8.

5.  The similarity index is 22% which is high for a reputed journal like Mathematics! It should be below 15%. Correct it.

6.      Some references (such as 8, 10 etc.) are not relevant to the work. Remove those and add more relevant and recent works.

Round 2

Reviewer 1 Report

None